# Enhancing fairness in disease prediction by optimizing multiple domain adversarial networks

**Bin Li**[1], **Xiaoqian Jiang**[2], **Kai Zhang**[2], **Arif O. Harmanci**[2], **Bradley Malin**[3], **Hongchang Gao**[1], **Xinghua Shi**[1,4]*, **for the Alzheimer's Disease Neuroimaging Initiative**[1‡]

**1** Department of Computer and Information Sciences, Temple University, Philadelphia, Pennsylvania, United States of America, **2** D. Bradley McWilliams School of Biomedical Informatics, The University of Texas Health Science Center at Houston, Houston, Texas, United States of America, **3** Department of Biomedical Informatics, Vanderbilt University, Nashville, Tennessee, United States of America, **4** Institute for Computational Molecular Science, Temple University, Philadelphia, Pennsylvania, United States of America

‡ Data used in preparation of this article were obtained from the Alzheimer's Disease Neuroimaging Initiative (ADNI) database (adni.loni.usc.edu). As such, the investigators within the ADNI contributed to the design and implementation of ADNI and/or provided data but did not participate in analysis or writing of this report. A complete listing of ADNI investigators can be found at: http://adni.loni.usc.edu/wp-content/uploads/how_to_apply/ADNI_Acknowledgement_List.pdf.
* mindyshi@temple.edu

**Data availability statement:** The ADNI dataset analysed in this study is available from the

## Abstract

Predictive models in biomedicine need to ensure equitable and reliable outcomes for the populations they are applied to. However, biases in AI models for medical predictions can lead to unfair treatment and widening disparities, underscoring the need for effective techniques to address these issues. However, current approaches struggle to simultaneously mitigate biases induced by multiple sensitive features in biomedical data. To enhance fairness, we introduce a framework based on a Multiple Domain Adversarial Neural Network (MDANN), which incorporates multiple adversarial components. In an MDANN, an adversarial module is applied to learn a fair pattern by negative gradients back-propagating across multiple sensitive features (i.e., the characteristics of patients that should not lead to a prediction outcome that may intentionally or unintentionally lead to disparities in clinical decisions). The MDANN applies loss functions based on the Area Under the Receiver Operating Characteristic Curve (AUC) to address the class imbalance, promoting equitable classification performance for minority groups (e.g., a subset of the population that is underrepresented or disadvantaged.) Moreover, we utilize pre-trained convolutional autoencoders (CAEs) to extract deep representations of data, aiming to enhance prediction accuracy and fairness. Combining these mechanisms, we mitigate multiple biases and disparities to provide reliable and equitable disease prediction. We empirically demonstrate that the MDANN approach leads to better accuracy and fairness in predicting disease progression using brain imaging data and mitigating multiple demographic biases for Alzheimer's Disease and Autism populations than other adversarial networks.

Alzheimer's Disease Neuroimaging Initiative (ADNI) repository (https://adni.loni.usc.edu/). The ABIDE II dataset used in this study is available at the ABIDE repository (https://fcon_1000.projects.nitrc.org/indi/abide/).

**Funding:** XJ is CPRIT Scholar in Cancer Research (RR180012), and he was supported in part by the Christopher Sarofim Family Professorship, UT Stars award, UTHealth startup, the National Institute of Health (NIH) under award numbers R01AG083039, U24LM013755 and U54HG012510, and the National Science Foundation (NSF) #2124789. This manuscript is partially supported by an NIH grant to XS (1R01NS140142-01).

**Competing interests:** The authors have declared that no competing interests exist.

## Author summary

In the realm of healthcare, the promise of personalized medicine through predictive modeling has been shadowed by the challenge of biases, which can skew outcomes and exacerbate disparities. In this study, we delve into developing unbiased predictive machine learning models that serve diverse populations fairly. We propose the Multiple Domain Adversarial Neural Network (MDANN), a framework that addresses the complex issue of fairness in biomedical disease prediction by simultaneously considering multiple sensitive attributes. By integrating adversarial components to learn unbiased patterns and employing a minimax loss function that focuses on the Area Under the Receiver Operating Characteristic Curve (AUC), we enhance classification performance for underrepresented groups. Our experimentation with brain imaging data for Alzheimer's Disease and Autism showcases MDANN's capability to improve accuracy and fairness in predictions, marking a significant advancement toward equitable healthcare outcomes. Through this initiative, we demonstrate the potential of MDANN in biomedical applications and underscore the importance of fairness in the development of AI models for healthcare, paving the way for future research in AI ethics and fairness.

## Introduction

Precision medicine, characterized by treatments and interventions tailored to individual genetic, environmental, and lifestyle factors, stands at the forefront of revolutionizing healthcare. This personalized approach promises enhanced effectiveness and fairness, potentially overcoming biases inherent in generalized healthcare practices. By leveraging unique patient profiles, precision medicine aims to provide equitable care across diverse populations, addressing disparities that may disadvantage specific groups.

The advent of artificial intelligence (AI), particularly deep learning (DL), in biomedical data analysis has heralded a new era of possibilities for enhancing healthcare outcomes [1]. Yet, the uncritical application of these technologies risks perpetuating, if not exacerbating, existing inequities within healthcare systems [2]. Biases embedded within data and algorithmic models can lead to disparate healthcare outcomes, underscoring the necessity of distinguishing between bias and fairness to ensure equitable patient care [3].

Mitigating biases in AI/DL models presents a formidable challenge, particularly in the context of biomedical data [4]. The identification and quantification of biases are crucial first steps toward developing fair AI systems. However, preventing these biases from influencing AI/DL applications in healthcare remains complex, given the propensity for biomedical datasets to reflect demographic imbalances, socioeconomic disparities, and varying healthcare practices [5]. Consequently, AI/DL predictions optimized on such data risk endangering minority groups through inaccurate or misleading outcomes.

To address these challenges, we propose the Multiple Domain Adversarial Neural Network (MDANN) framework, which directly targets the mitigation of biases induced by multiple sensitive features such as race, gender, or age. Current bias-mitigation approaches [6–13] often focus on one sensitive attribute at a time, but this can overlook the complex interactions between multiple demographic factors. MDANN overcomes this by employing adversarial components designed to minimize the influence of these sensitive features across the learning process, ensuring the model learns representations that are invariant to such biases. Through this adversarial learning approach, the model can mitigate bias from several sensitive attributes simultaneously, promoting fairness across various demographic groups.

In this paper, we introduce the notion of a Multiple Domain Adversarial Neural Network (MDANN), which is designed to simultaneously mitigate biases from multiple sensitive features, in contrast to traditional approaches that may tackle biases in a sequential or isolated manner. The proposed MDANN framework provides several notable contributions to bias mitigation and fairness enhancement in biomedical data analysis.

**First**, MDANN learns a common and rich data representation for all patients in the training data. This representation is used as the basis for training a set of adversarial neural network models that learn to train models that satisfy fairness requirements in predicted disease risk. We demonstrate that our approach can flexibly balance the bias, accuracy, and efficient tradeoff. Additionally, we show that the convolutional autoencoder (CAE) is an effective feature extractor, contributing to enhanced prediction fairness.

**Second**, we employ an AUC-induced minimax loss function that takes into account the Area Under the Receiver Operating Characteristic Curve (AUC) score, in contrast to the conventional accuracy-induced loss, such as cross-entropy. As we empirically illustrate, the minimax loss function outperformed the standard cross-entropy loss in handling imbalanced data, leading to improved classification performance and fairness in predictions for the minority class.

**Finally**, we investigate the impact of adversarial modules on prediction performance in the context of bias mitigation using the MDANN. By addressing multiple sensitive features, we demonstrate that the introduction of adversarial components effectively enhances fairness in disease prediction on ADNI and ABIDE II datasets.

## Materials and methods

In this section, we introduce the MDANN approach for fairness enhancement in disease prediction. The source code is available at a Github repository [14]. Fig 1 depicts the MDANN architecture, which is tailored to the unique demand for fairness enhancement and is composed of three fundamental and synergistic modules (shown in bold): **1) Feature Extractor:** This module is a pre-trained convolutional autoencoder designed to delve into the complex structure of biomedical data, extracting deep and meaningful representations that serve as the foundation for subsequent calculations; **2) Adversarial Module:** This module consists of a set of adversarial neural networks to learn fair patterns across various domains. The gradients of the module are averaged and reversed by the gradient reversal layer and **3) AUC-induced Minimax Loss Function:** A carefully designed loss function that focuses on the AUC, providing a robust and sensible metric for imbalanced data classification, and underscoring our nuanced understanding of the challenges posed by imbalanced biomedical data.

In composition, these modules form an integrated and end-to-end learning network, each contributing unique strengths and working in harmony to extract informative representations, mitigate multiple biases, and promote fairness against label imbalance problems. In the following sections, we introduce each module, detailing its design and functionality within the MDANN framework.

### Feature extractor

The feature extraction process in this study employs a pre-trained convolutional autoencoder (CAE) model, consisting of convolutional layers serving as the encoder and deconvolutional layers acting as the decoder. The CAE model is designed to acquire underlying patterns in the data to generate images resembling the original input images. Within the context of the MDANN framework, deep representations of imaging samples are extracted from the last convolutional layer (bottleneck) of the model. This approach facilitates the transformation

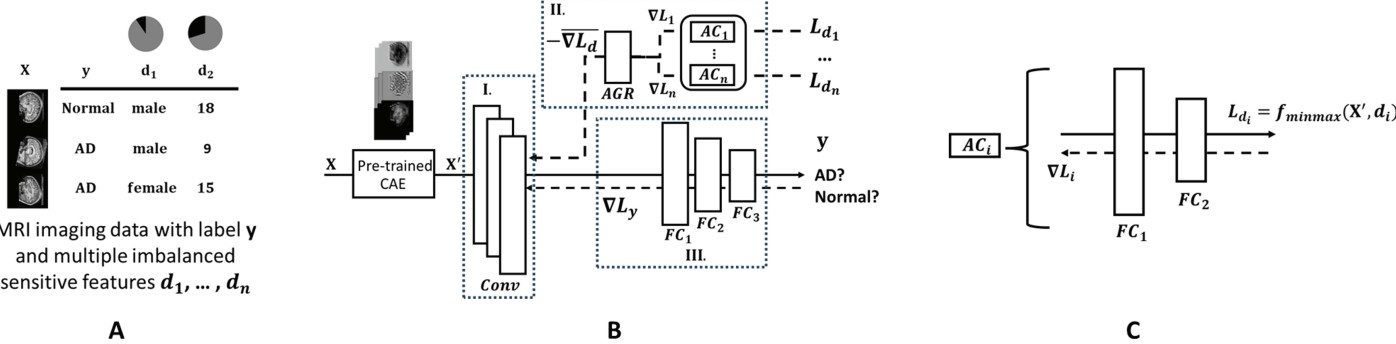

**Fig 1. The scheme of MDANN.** A. An illustration of training data. B. The model architecture of MDANN. The MDANN incorporates three modules including a feature extractor (I.), an adversarial module (II.), and a predictor (III.). The feature extractor (I.) is used to extract the embedding content $X'$ as a deep representation from imaging samples **X**. The adversarial module (II.) contains several adversarial components (ACs) which are used to learn distributions of multiple sensitive features simultaneously and pass back the negative gradients through an averaged gradient reversal layer (AGR). The predictor (III.) consists of three fully connected layers for classification on main labels $y$. C. The architecture of adversarial component (AC). Each AC is composed of two fully connected (FC) layers acting as classifiers. The minimax loss has been calculated based on embeddings $X'$ and specific sensitive feature $d_i$ via forward propagation. Gradients $\nabla L_i$ were calculated and sent back for average reversal.

of 2D images into high-dimensional vectors in latent space through multiple convolutional calculations.

While convolutional autoencoders can provide high-quality representations of the data, they do not guarantee explicit interpretability of the latent features. The latent space captures abstract patterns that may not align directly with human-interpretable features. To mitigate this concern, we have implemented multiple adversarial components that are specifically tasked with learning representations that are invariant to sensitive attributes like race, gender, and age. Although CAEs excel at extracting robust features, these adversarial layers ensure that the representations focus on relevant predictive features while minimizing biases from sensitive attributes.

CAE is often considered more effective at decreasing biases than other feature extractors due to its ability to learn more robust and representative features from data. Its convolutional nature enables it to capture spatial hierarchies in data, which can be crucial for reducing biases, particularly in image-related tasks [15]. By utilizing the pre-trained CAE, our method endeavors to enhance the fidelity and informativeness of the extracted representations, ultimately contributing to the reduction of biases and promoting fairness in the subsequent stages of predictive modeling.

### Adversarial module

The adversarial module is implemented through the incorporation of several adversarial components (AC), as shown in Fig 1C, each consisting of two linear layers that act as predictors. These components are integrated with the feature extractor by employing a gradient reversal layer, which exerts its influence during backpropagation-based training. More specifically, the gradient reversal layer operates by multiplying the averaged negative gradient, thereby facilitating the adversarial alignment of feature distributions across biased groups. The primary objective of the adversarial module is to achieve robust adversarial learning, wherein the feature distributions pertaining to different sensitive attributes are as indistinguishable as possible for each component.

A significant contribution of our approach lies in the simultaneous mitigation of multiple biases through the use of multiple adversarial networks. By incorporating multiple adversarial components, each catering to a distinct sensitive attribute, the model is endowed with the capability to learn specialized representations that are invariant to the influence of sensitive attributes. Consequently, our approach reduces the likelihood of biased predictions based on these attributes, fostering fairness and equitable predictions across diverse groups.

Our approach is bolstered by the utilization of a gradient reversal layer, which supports robust adversarial learning. By aligning feature distributions across biased groups, the model is encouraged to minimize discrepancies in its predictions between majority and minority groups. This mechanism ensures that the model incurs substantial penalties for mispredictions on a minority group, thereby promoting fairness and equitable outcomes. The combination of specialized representations for sensitive attributes and the robust adversarial learning mechanism contributes to the model's ability to reduce biased predictions and achieve equitable and reliable predictive performance across diverse groups, making it a promising advancement in the field of biomedical data analysis.

## Optimizer on imbalanced data with a minimax loss

The conventional loss function based on optimizing accuracy is a widely employed optimization function for binary and multi-class classification tasks. However, its effectiveness is limited when dealing with imbalanced data, where skewed class distributions pose significant challenges. In such scenarios, the model often stagnates during training, consistently predicting all test samples as the majority category. This phenomenon leads to artificially high accuracy but results in very low precision or recall for the minority class, highlighting the inadequacy of conventional loss functions in addressing label imbalance. Therefore, more robust optimization strategies are desirable to achieve fair and equitable predictive outcomes even when data is imbalanced.

To tackle the issue of label imbalance in biomedical data analysis, recent studies have focused on optimizing the Area Under the Receiver Operating Characteristic Curve (AUC), a sensible and robust metric for imbalanced-based data classification tasks [16–18]. The distinctive property of AUC lies in its aggregation across different threshold values for binary prediction, decoupling the issues of threshold setting from the model's predictive power. Moreover, AUC considers both precision and recall, providing a more comprehensive evaluation of a model's performance, particularly in the context of imbalanced data.

While weighted cross-entropy loss has been proposed as a solution to handle class imbalance, it suffers from certain limitations. Weighted cross-entropy adjusts the loss function by assigning higher weights to minority samples, but it does not inherently account for the trade-off between false positive and false negative rates, which are critical in imbalanced settings. In contrast, the AUC-based minimax loss explicitly incorporates these trade-offs, making it more suitable for improving performance in the minority class.

Given these advantages, we employ the minimax loss function proposed by Ying et al. [18] in the adversarial module of our MDANN framework. This loss function directly optimizes the AUC score, allowing the model to focus on enhancing performance for the minority class, which is crucial for achieving fairness in predictions. The formulation of the minimax loss is as follows:

$$
\begin{aligned}
minimax\,\mathfrak{L}(w, \mu_1, \mu_2, \theta; a, b) = {} & (1 - p)(f(w; a) - \mu_1)^2 \mathbb{1}(b = 1) \\
& + p(f(w; a) - \mu_2)^2 \mathbb{1}(b = 0) - p(1 - p)\theta^2 \\
& + 2(1 + \theta)(pf(w; a))\mathbb{1}(b = 0) - (1 - p)f(w; a)\mathbb{1}(b = 1)) \qquad (1)
\end{aligned}
$$

The goal of this function is to find the model weights and parameters that minimize the maximum loss over all samples, thereby improving the model's performance on the minority class and enhancing fairness in the model's predictions. Moreover, this loss function inherently considers sample weighting, in which terms are weighted by $(1-p)$ and $p$, the proportions of negative and positive samples, respectively. By incorporating the sample weighting term $p$, this loss function not only seeks to optimize overall classification performance but also ensures that the imbalance between classes is directly accounted for during optimization.

## Experimental setup

All experiments were conducted on a cluster server with four NVIDIA RTX A5000 GPUs. We evaluated the performance of the MDANN with a varying number of adversarial components. A grid-search method was applied to fine-tune the parameters in experiments.

To ensure the reliability of our results, each experiment was run five times. Running multiple iterations allows us to account for variability in model performance due to randomness in initialization and training processes, and it helps ensure that the observed outcomes are robust and not the result of any particular random seed. This is a common practice in machine learning research to provide more stable and generalizable results, particularly balancing the need for computational feasibility and statistical robustness.

### Datasets

We focus on two biomedical datasets. The first dataset is from Autism Brain Imaging Data Exchange II (ABIDE II) [19], which is a repository that aggregates and openly shares resting-state functional magnetic resonance imaging (R-fMRI) data sets with corresponding structural MRI and phenotypic information with autism spectrum disorder (ASD) and typical controls (TC). We collected MRI imaging as well as phenotypic data from 212 individuals, with 110 ASDs and 102 TCs. The second dataset is from the Alzheimer's Disease Neuroimaging Initiative (ADNI) database [20,21]. The ADNI was launched in 2003 as a public-private partnership, led by Principal Investigator Michael W. Weiner, MD. The primary goal of ADNI has been to test whether serial magnetic resonance imaging (MRI), positron emission tomography (PET), other biological markers, and clinical and neuropsychological assessment can be combined to measure the progression of mild cognitive impairment (MCI) and early Alzheimer's disease (AD). We collected 670 MRI imaging data with demographic information labeled as three classes: 350 normal cognitive (NC) samples, 152 MCI samples, and 168 AD samples.

The labeling strategy employed in this study is designed to facilitate binary classification. In the autism dataset, ASDs are labeled as 1, and TCs are labeled as 0. This labeling reflects the objective to distinguish between individuals with ASD and typical controls. In the ADNI dataset, NC is labeled as 0, and both MCI and AD are collectively labeled as 1. This approach enables the differentiation between normal cognitive function and conditions indicative of cognitive impairment, including both mild cognitive impairment and Alzheimer's disease.

To ensure robust evaluation and reproducibility, we employed a 5-fold stratified cross-validation procedure. For each dataset, the data was partitioned into training (80%) and testing (20%) sets. The stratification process guarantees that each fold maintains the same proportion of classes and sensitive attributes, ensuring fair and unbiased evaluation of model performance. In the context of demographic data, a majority label refers to a specific category or group that may have advantages in group size or are favored in a particular context. These labels often correspond to characteristics or attributes that are considered "normative" or "majority" within a given society or dataset. Therefore, the majority and minority groups

for the two datasets are defined as shown in Fig 3. For the Autism dataset, the majority group corresponds to the right-handedness and male subpopulation. For the ADNI dataset, the majority group corresponds to individuals over 78 years old, with more than 18 years of education, and are right-handed. Thus, MDANN with two and three adversarial components were designed for the autism and ADNI datasets, respectively.

## Evaluation metrics

We apply two metrics to analyze and address fairness that would be present in the training data. First, we use AUC for binary classification problems, which measures the ability to distinguish between false positive and negative rates. It is important to note, however, that while AUC is a valuable measure of a model's performance, it does not inherently provide a measure of fairness. Thus, to measure fairness, it is crucial for adversarial components to distinguish between the minority (represented as 1) and majority (represented as 0) attributes, which makes AUC a necessary metric for MDANN. In this context, the majority group refers to the demographic group with a higher representation or prevalence in the dataset. For instance, if the dataset contains more males than females, males would be the majority group. Similarly, if right-handed individuals outnumber left-handed individuals, the right-handed group is considered as the majority group. Conversely, the minority group refers to the demographic group with fewer samples in the dataset. These definitions apply to both sex (male vs. female) and handedness (right-handed vs. left-handed) or any other sensitive attribute under consideration.

Assume that for some dataset, we know the ground truth outcomes $Y \in \{0, 1\}$, and refer to $\hat{Y} = 1$ as the favored outcome, assuming it represents the more desirable of the two possible results. We employ the Disparate Impact (DI) and Equal Opportunity (EO) [22] and True Negative Rate Disparity (TNRD) as fairness evaluation metrics. DI compares the proportion of individuals that receive a positive output for the majority and minority groups. DI is computed as follows,

$$DI = \frac{Sr_{minority}}{Sr_{majority}}, SR = Pr(\hat{Y} = 1 | D = Group) \tag{2}$$

where $Sr_{minority}$ and $Sr_{majority}$ are the selection rates for the minority and majority groups, respectively. $Pr(Y \hat{=} 1 | D = minority)$ is the proportion of the minority group that received the positive outcome and $Pr(Y \hat{=} 1 | D = majority)$ is the proportion of the majority group that received the positive outcome. The DI value ranges from 0 to infinity, with 1 indicating no disparate impact, meaning that the model treats all groups equally in terms of favorable outcomes. Values greater than 1 indicate a positive disparate impact, suggesting that the majority group receives more favorable outcomes than the minority group, which may indicate bias in favor of the majority group. Conversely, values lower than 1 indicate a negative disparate impact, suggesting that the majority group receives fewer favorable outcomes than the minority group, which may indicate a bias against the majority group.

EO is another fairness metric commonly used in machine learning to assess the disparate impact of a model's predictions across different groups. It specifically focuses on equalizing true positive rates across majority and minority groups. The calculation of EO is as follows,

$$
\begin{aligned}
EO &= |TPR_{minority} - TPR_{majority}| \\
&= |Pr(\hat{Y} = 1 | Y = 1, D = minority) - Pr(\hat{Y} = 1 | Y = 1, D = majority)|
\end{aligned}
\tag{3}
$$

EO represents the disparity in true positive rates between the two groups. If the EO is close to 0, it indicates that the model is providing similar opportunities for true positive predictions across the specified groups. A positive value suggests a higher true positive rate for the minority group, while a negative value suggests a higher true positive rate for the majority group.

TNRD evaluates the disparity in True Negative Rates (TNR) across the majority and minority groups. It is calculated as:

$$
\begin{aligned}
TNRD &= |TNR_{minority} - TNR_{majority}| \\
&= |Pr(\hat{Y} = 0|Y = 0, D = minority) - Pr(\hat{Y} = 0|Y = 0, D = majority)|
\end{aligned} \tag{4}
$$

TNRD evaluates the disparity in TNR between the majority and minority groups. It complements the EO, which focuses on the TPR, by addressing fairness in the negative predictions ($Y = 0$). TNRD ensures that the model does not exhibit biased behavior when predicting negative outcomes, which is especially important in sensitive applications like disease detection, where both false positives and false negatives can have serious consequences.

## Results

### Fairness enhancement achieved by the feature extractor

The input values were fed to a pre-trained convolutional autoencoder (CAE), which is used to extract deep representations from the embedding layer. By learning to reconstruct the input, the autoencoder encourages the embedding to capture the most important and informative aspects of the data, while minimizing the impact of biased information. In MDANN, the CAE module contains several convolutional layers as the image encoder and the same number of deconvolution layers as the decoder. Each layer employs a 5×5 kernel with a step size of 2 for its calculations. CAEs are particularly effective for image data, as they can capture hierarchical feature representations from lower to higher levels of abstraction. This hierarchical approach allows the CAE to model both fine-grained local features and more abstract global structures within the data, thereby improving its ability to extract deep representations that are crucial for fair prediction models [23,24]. Table 1 shows the prediction result of MDANN using the pre-trained CAE with different sizes of encoders and decoders. The embedding size represents the dimensions of the learned latent space, where smaller embedding sizes indicate higher compression of information. For instance, in CAE-1, the embedding size of 64×64×128 indicates that the data has been compressed into a latent space with spatial dimensions of 64×64 and 128 feature maps. In contrast, CAE-2 with 32×32×256 provides a more compact representation, compressing the input further while increasing the number of feature maps to 256. The row labeled "N/A" in Table 1 indicates that no CAE module was applied to the data. In this case, the original data was used directly for training without any feature extraction or dimensionality reduction. This serves as a baseline comparison to highlight the benefits of deep feature extraction provided by the CAE. We chose three specific embedding sizes to provide a balance between the amount of compression and the richness of the extracted features. These choices show that an optimal balance between embedding size and the number of feature maps often leads to better generalization and fairness in predictive tasks. Prior studies [25,26] have similarly shown that striking the right balance between spatial dimensions and feature richness can improve both predictive accuracy and fairness, particularly in complex data tasks. To assess the image quality generated by the CAE modules, we adopted the Fréchet inception distance (FID) as a metric [27]. A lower FID

**Table 1. Performance of MDANN with different CAE modules.**

| Module | Embedding Size | Convolutional layers | FID score | F1 score | Averaged DI |
|---|---|---|---|---|---|
| N/A | N/A | N/A | N/A | 0.61 | 0.68 |
| CAE-1 | 64×64×128 | 1 | 556.74 | 0.61 | 0.67 |
| **CAE-2** | **32×32×256** | **2** | **490.98** | **0.72** | **0.77** |
| CAE-3 | 16×16×512 | 3 | 555.54 | 0.67 | 0.72 |

score indicates an improved distribution of generated images. Experiments show that CAE-2 could reach the maximum F1 score of 0.72 and averaged DI of 0.77 for sensitive attributes for AD prediction. By contrast, MDANN achieves only 0.61 F1 score and 0.68 averaged DI when utilizing original data without any feature extraction (as depicted in the first row of Table 1).

An empirical approach was applied to set the experimental design. It is natural to believe that adding more embedding layers might enable better reconstruction, and thus, preserve more details about the input data. However, our analysis indicates that this does not always translate into better performance for specific downstream tasks. When the number of convolutional layers is increased to 3 (CAE-3), we observe a lower FID score than that achieved by CAE-2. Thus, the CAE-2 was selected as the feature extractor for the following experiments. Fig 2 illustrates the generative images of CAE-2 during a 100-epoch training process for a single sample image. It is evident that the output image converges satisfactorily over the course of these epochs. Therefore, to obtain a stable deep representation, CAE training was extended to 200 epochs.

To assess potential biases in the embeddings extracted by the CAE, we analyzed the correlations between sensitive attributes (e.g., demographic traits such as sex and handedness) and the generated embeddings. The correlations were computed using the Pearson correlation coefficient, which quantifies the linear relationship between sensitive attributes (binary values) and embedding features. The computation process is as follows:

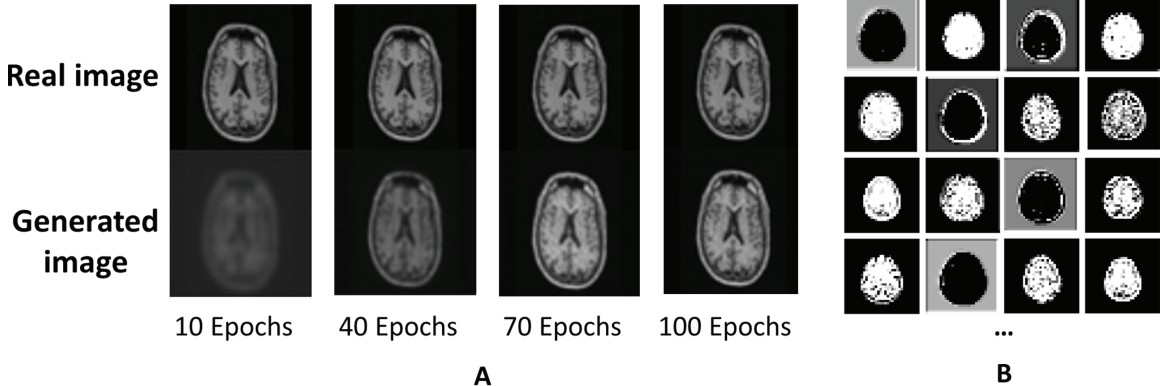

**Fig 2. The training results for the CAE-2 module.** A. Generated images from CAE-2 with 10, 40, 70 and 100 epochs for one real sample image. B. The averaged Frechet Inception Distance (FID) between real images and generated images for evaluating the quality of generation during the training process.

- Embedding Feature Aggregation: For each individual, embeddings generated by the CAE modules (CAE-1, CAE-2, and CAE-3) or raw features (Baseline without CAE) were aggregated across dimensions. This produced a single scalar value per individual, representing the overall embedding.
- Attribute-Feature Mapping: Sensitive attributes (e.g., sex or handedness) were mapped as binary values (e.g., 0 for male and 1 for female) to each individual.
- Correlation Computation: Using the aggregated embedding values and binary-sensitive attributes, the Pearson correlation coefficient was calculated to measure the strength and direction of the relationship.

The results, presented in Table 2, show that embeddings generated by CAE modules combined with adversarial processing significantly reduce their correlations with sensitive attributes and thus effectively mitigate biases from these sensitive attributes. For instance, the correlation between embeddings and sex decreased from 0.45 (Baseline without CAE) to 0.20 (CAE-2), while the correlation with handedness dropped from 0.52 to 0.18. These findings demonstrate the effectiveness of combining adversarial processing with CAE to mitigate bias within the learned embeddings.

### Improved optimization strategies to handle label imbalance

Since each sensitive label for training is imbalanced, (i.e., $\frac{N_{majority}}{N_{minority}} > 2$) we investigated if all ADs collaboratively optimize the loss function to learn the best model parameter against label imbalance. Fig 3 shows the specific label distribution for each sensitive feature in the

**Table 2. Correlation analysis between sensitive attributes and embeddings across different CAE modules.**

| Module | Correlation with Sex | Correlation with Handedness |
|---|---|---|
| Baseline (No CAE) | 0.45 | 0.52 |
| CAE-1 | 0.38 | 0.45 |
| **CAE-2** | **0.20** | **0.18** |
| CAE-3 | 0.25 | 0.22 |

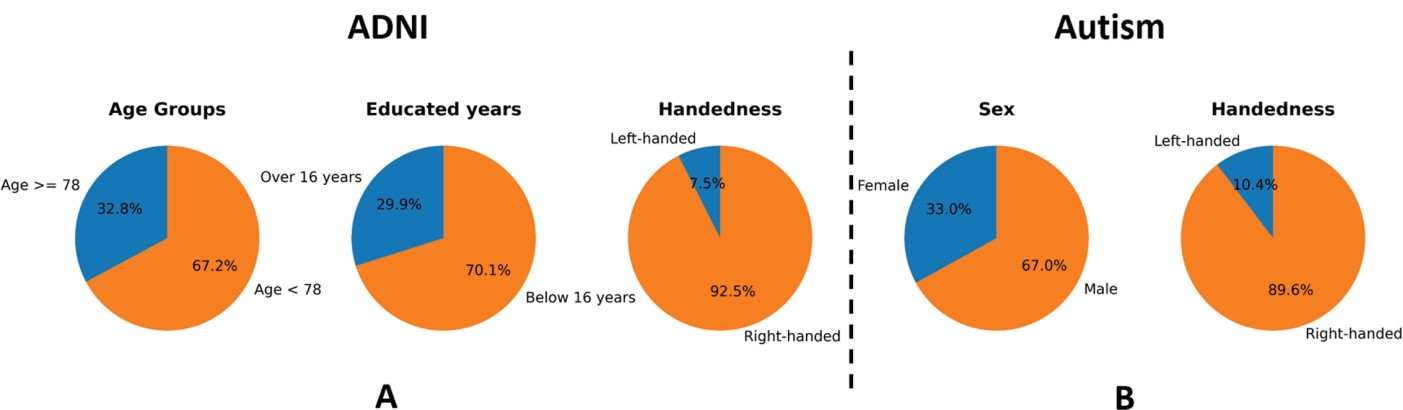

**Fig 3. The demographic Distribution in ADNI and Autism datasets.** A. The distribution of Age Groups, Handedness, and Education in the ADNI dataset. The left chart categorizes individuals into two age groups, the middle chart shows the distribution of left-handed and right-handed individuals, and the right chart divides the population into two education levels. B. The distribution of Handedness and Sex in the Autism dataset. The leftmost chart illustrates the proportion of left-handed and right-handed individuals, while the rightmost chart delineates the gender distribution. The majority groups are represented in orange color and minority groups are in blue.

two datasets. It can be observed that the number of majority instances is greater than that of minority instances. Generally, the label-imbalance problem could lead to a biased prediction. Specifically, the machine learning models are prone to predict the minority class as the majority class. We kept hyperparameters and model architecture the same except for the loss function. Then, we tested the performance of two prediction tasks with two different loss functions (minimax loss and sample-weighted cross-entropy loss function as a comparison) during the training process, as shown in Fig 4. Results of the prediction utilizing different losses and optimizers are shown as two rows, accuracy (ACC)-induced weighted Cross-Entropy Loss and AUC-induced Minimax Loss. Each row contains four experimental settings, showing the classification performance (ACC) for each sensitive feature and the global AUC score for the predictor. AUC results show that a minimax loss does improve the classification performance while using cross-entropy loss could only achieve around 0.6 AUC score, tending to be less distinguishable between minority (represented as 1) and majority (represented as 0) attributes. Notably, the prediction accuracy for each of the two categories for handedness converges to 0.6 because it did not learn any relationship between the features and labels likely due to the high-imbalanced data ($\frac{N_{majority}}{N_{minority}}$ = 12.33). The second row illustrates that the weighted Minimax loss was able to overcome the label-imbalanced problem, where the AUC score increased to 0.8. This demonstrates that employing a minimax loss significantly increases the accuracy for minority classes. The predictor achieved an accuracy of 0.8 for patients who were left-handed, but only achieved 0.6 when using a classical cross-entropy loss.

## Bias mitigation via multiple components in the adversarial module

Next, we investigated the impact of the number of adversarial components on prediction performance in the context of fairness enhancement. To do so, we trained the MDANN while simultaneously addressing multiple sensitive features for AD prediction tasks. We systematically compared three variations of the MDANN, each with a different number of adversarial modules. Fig 5 depicts the predictive performance. Here, panels A and B illustrate the accuracy and AUC score of predictors employing different numbers of adversarial components while incorporating cross-entropy loss into the MDANN. The red line represents the baseline,

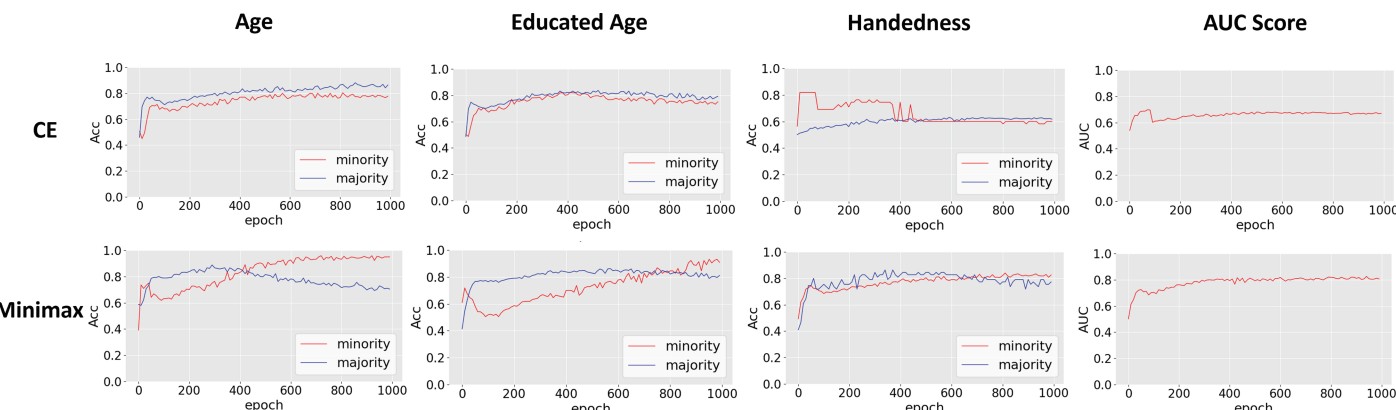

**Fig 4. Prediction results of MDANN with respect to different loss functions.** Rows represent the prediction metrics with different loss functions: weighted cross-entropy loss (CE) and minimax loss. The first three columns illustrate the accuracy of the predictor for each of the sensitive features in the ADNI dataset. The final column depicts the AUC score of the predictor.

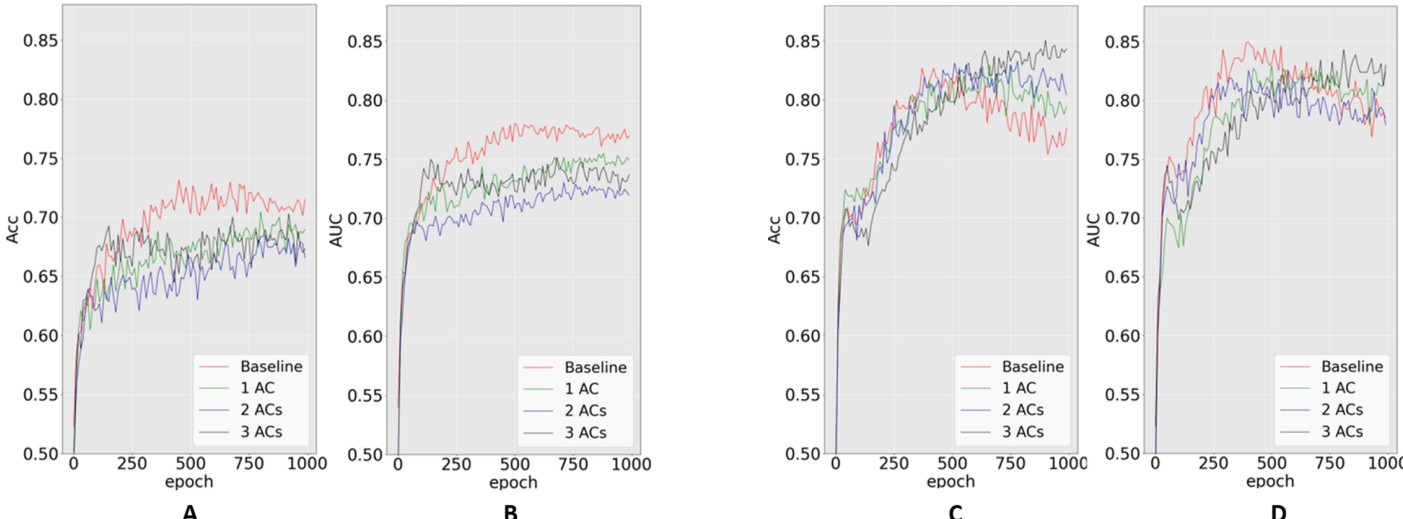

**Fig 5. Validation accuracy (Acc) and AUC score for ADNI disease prediction under two scenarios.** Accuracy (A) and AUC (B) of MDANN trained with cross-entropy loss. Accuracy (C) and AUC (D) of MDANN trained with minimax loss. Three biased attributes: age, handedness, and education years, were used as three embedded adversarial components (ACs) for bias mitigation. The baseline was the regular machine learning model trained without any AC. All images have been fed to a pre-trained CAE with two convolutional layers (with deep representations size of 32×32×256).

which corresponds to training without an adversarial module. When optimizing the model using cross-entropy loss, we observed that increasing the number of adversarial components led to a reduction in prediction performance, as evidenced by diminished accuracy and AUC scores. Specifically, the lowest accuracy of 0.65 was achieved for disease prediction when the MDANN simultaneously attempted to address four attributes. This result is attributed to the inclusion of more imbalanced features during the training process, which the cross-entropy loss cannot effectively address. Additionally, panels C and D illustrate the accuracy and AUC score when employing minimax loss for training. It can be seen that during the traditional training (baseline), the accuracy initially reached a peak and then declined, eventually stabilizing at an accuracy of 0.76 over 1000 epochs. This behavior reflects the outcome of the AUC-induced minimax loss, given its objective of maximizing the AUC and, thereby, precludes a guarantee that accuracy is consistently optimized. Fortunately, by increasing the number of adversarial components, fairness was significantly improved without sacrificing classification performance. As the number of imbalanced labels increased, the number of negative gradients propagated backward also increased, leading to a reduction in the convergence speed. Consequently, employing more adversarial components overcame this challenge and effectively combined the advantages of the loss function, ultimately achieving the highest accuracy of 0.85 for AD prediction.

We provide a comprehensive evaluation of fairness (Table 3) using various metrics across different sensitive attributes (age, educated years, and handedness). These results depict evaluation metrics including Accuracy, AUC, F1, DI, EO, and TNRD for each sensitive feature under different configurations (Baseline, 1 AC, 2 ACs, and 3 ACs), with CAE-2 as the feature extractor. The results show that increasing the number of adversarial components leads to consistent improvements in both performance and fairness metrics across all sensitive attributes. Specifically, we observe improvements in F1 alongside rising DI values (approaching 0.88) and decreasing EO values (as low as 0.06 for handedness) when employing three

**Table 3. Fairness Metrics with Standard Deviations Across Sensitive Attributes ($A_1$: Age, $A_2$: Educated Years, $A_3$: Handedness) with Varying Numbers of Adversarial Components (ACs).**

| Configuration | ACC | AUC | F1 | DI | | | EO | | | TNRD | | |
|---|---|---|---|---|---|---|---|---|---|---|---|---|
| | | | | $A_1$ | $A_2$ | $A_3$ | $A_1$ | $A_2$ | $A_3$ | $A_1$ | $A_2$ | $A_3$ |
| Baseline | 0.69 | 0.71 | 0.70 | 0.61 | 0.60 | 0.63 | 0.19 | 0.21 | 0.19 | 0.18 | 0.22 | 0.20 |
| | ±0.02 | ±0.02 | ±0.02 | ±0.04 | ±0.05 | ±0.04 | ±0.03 | ±0.04 | ±0.03 | ±0.03 | ±0.04 | ±0.03 |
| 1 AC | 0.74 | 0.77 | 0.75 | 0.65 | 0.68 | 0.68 | 0.15 | 0.16 | 0.14 | 0.15 | 0.17 | 0.16 |
| | ±0.03 | ±0.03 | ±0.02 | ±0.03 | ±0.04 | ±0.03 | ±0.02 | ±0.03 | ±0.02 | ±0.02 | ±0.03 | ±0.02 |
| 2 ACs | 0.78 | 0.81 | 0.80 | 0.71 | 0.72 | 0.73 | 0.10 | 0.11 | 0.09 | 0.10 | 0.12 | 0.09 |
| | ±0.02 | ±0.02 | ±0.01 | ±0.02 | ±0.03 | ±0.02 | ±0.01 | ±0.02 | ±0.01 | ±0.01 | ±0.02 | ±0.01 |
| 3 ACs | 0.83 | 0.85 | 0.85 | 0.76 | 0.77 | 0.78 | 0.05 | 0.06 | 0.04 | 0.05 | 0.07 | 0.04 |
| | ±0.01 | ±0.01 | ±0.01 | ±0.02 | ±0.02 | ±0.02 | ±0.01 | ±0.01 | ±0.01 | ±0.01 | ±0.01 | ±0.01 |

adversarial components. Likewise, TNRD also decreases across all sensitive attributes, indicating reduced disparities in true negative rates. These findings underscore the efficacy of MDANN in mitigating biases and enhancing fairness in predictive tasks, particularly for imbalanced biomedical datasets.

## Combined mechanisms for fairness enhancement

The previous sets of experiments considered how the feature extractor, minimax loss, and adversarial module influenced the results independently. Thus, we next considered a more comprehensive evaluation of these components within the MDANN framework. We analyzed three fair-enhancing mechanisms applied to biomedical datasets, and we present their efficacy as modules in the MDANN. To do so, we tested the MDANN while simultaneously addressing multiple sensitive features for Autism prediction tasks. A comparison was conducted across various approaches, including two variants of MDANN with differing numbers of adversarial components, and three other methods, synthetic minority over-sampling technique (SMOTE) with sampling strategy of 0.5 (minority class will be 50% the size of the majority class after resampling), adversarial de-biasing network (AD-Net) [28], and domain adaptive neural network (DANN) [29]. The examination encompassed two sensitive features: sex and handedness.

Fig 6 depicts the performance of different models in mitigating biases and enhancing fairness in predictions. For sex, AD-Net demonstrated the lowest performance with an accuracy of 0.48, AUC of 0.50, F1 of 0.46, DI of 0.58, EO of 0.22, and TNRD of 0.26, reflecting poor predictive performance and significant bias. DANN showed moderate improvement, achieving an accuracy of 0.62, AUC of 0.63, F1 of 0.62, DI of 0.67, EO of 0.19, and TNRD of 0.21. SMOTE performed better in fairness metrics (DI: 0.71, EO: 0.18, TNRD: 0.20) but still fell short in overall predictive performance (accuracy: 0.62, AUC: 0.62, F1: 0.59). By contrast, MDANN with 2 adversarial components exhibited superior performance and fairness, achieving an accuracy of 0.71, AUC of 0.73, F1 of 0.72, DI of 0.84, EO of 0.12, and TNRD of 0.11.

A similar pattern was observed for handedness. AD-Net yielded the weakest results, with an accuracy of 0.49, AUC of 0.53, F1 of 0.48, DI of 0.62, EO of 0.23, and TNRD of 0.28. Although SMOTE achieved modest improvements in fairness (DI: 0.75, EO: 0.19, TNRD: 0.22), it was still outperformed by DANN in overall balance of metrics. DANN further improved fairness (DI: 0.76, EO: 0.17, TNRD: 0.18) and achieved an accuracy of 0.65, AUC of 0.63, and F1 of 0.63. Nevertheless, it was surpassed by MDANN with 2 adversarial components, which provided the best trade-off between predictive performance and fairness, with an accuracy of 0.70, AUC of 0.73, F1 of 0.71, DI of 0.84, EO of 0.12, and TNRD of 0.12.

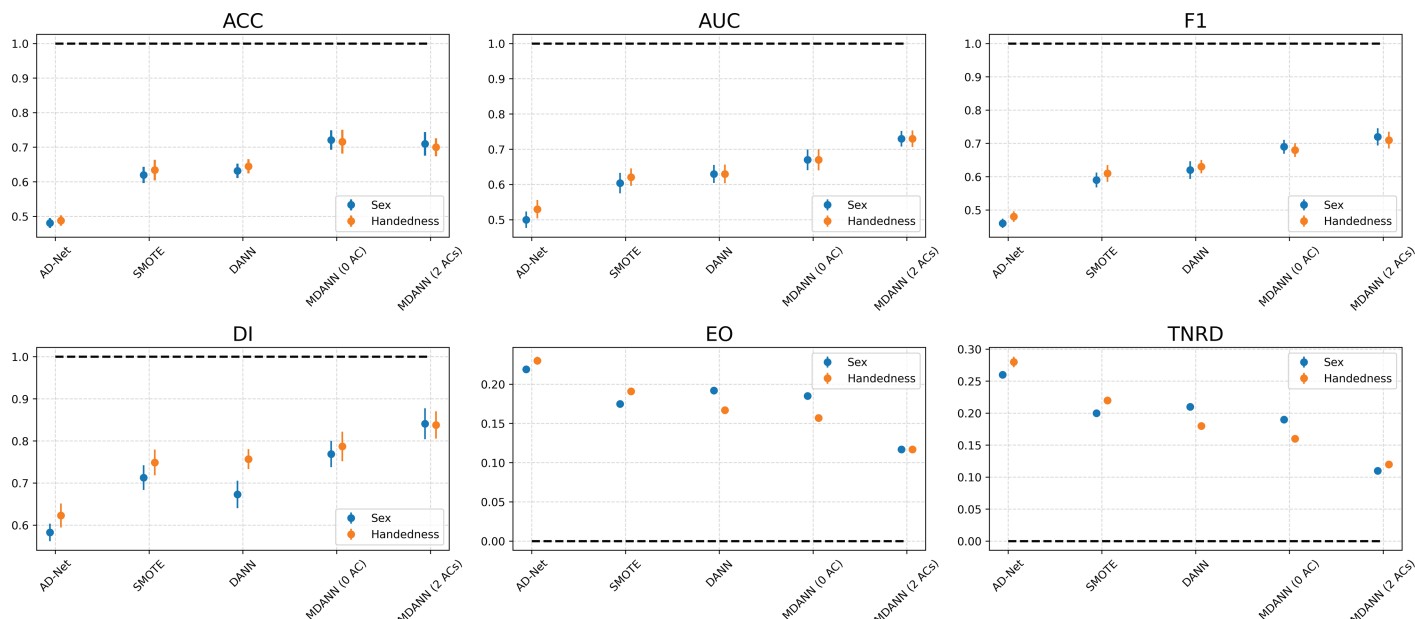

**Fig 6. Comparative performance of different methods of bias mitigation on sex and handedness.** This analysis depicts averaged results with four methods: 1) AD-Net, 2) DANN, 3) MDANN with 0 AC, and 4) MDANN with 2 ACs. The analysis considered two sensitive features where sex is shown in blue, and handedness is depicted in orange. The upper row depicts the predictive performance of ACC (top-left), AUC (top-middle), and F1-score (top-right). The lower row depicts the fairness performance of DI (bottom-left) and EO (bottom-middle), and TNRD (bottom-right). There are horizontal black dashed lines at y=1 and y=0, which serve as a reference. A DI closer to 1 and EO, TNRD closer to 0 indicate enhanced fairness. The results highlight the superior performance of MDANN, particularly the variation with 2 ACs, in terms of both predictive accuracy and fairness for Autism classification tasks.

Overall, these results demonstrate that MDANN with 2 adversarial components achieves the best balance among accuracy, AUC, F1, and fairness (DI, EO, TNRD) for both sex and handedness in Autism classification tasks.

## Conclusion

In this study, we introduced an MDANN framework, which incorporates three modules to perform fair prediction on biomedical imaging datasets. Our empirical findings highlight the efficacy of adversarial modules in the MDANN framework, effectively mitigating biases and promoting fairness by addressing multiple sensitive features. Additionally, the utilization of AUC-based minimax loss functions demonstrates their superior handling of label imbalance, leading to improved classification performance for the minority class. Furthermore, we showcase the potential of deep representations extracted from a pre-trained convolutional autoencoder, resulting in enhanced prediction accuracy and fairness in AD and Autism prediction. The investigations underscore the promising potential of MDANN as a method for fairness enhancement, particularly in scenarios where the sensitive attributes of sex and handedness are of concern, simultaneously. Our experimental results provide intuition into the reasons for MDANN's superiority, reflecting its potential as a valuable method for enhancing both predictive performance and fairness. MDANN's demonstrated effectiveness in improving fairness for minority classes indicates its potential to generalize well across different disease prediction tasks, particularly those that involve demographic or feature-based biases.

However, this study focuses on marginal fairness, which evaluates fairness for individual sensitive attributes independently, rather than intersectional fairness, which considers overlapping sensitive attributes, such as sex and ethnicity or age and socioeconomic status.

While MDANN is designed to mitigate biases for multiple sensitive attributes simultaneously, its adversarial modules operate independently and do not account for potential interactions between attributes. We acknowledge that marginal fairness does not guarantee fairness for intersectional subgroups defined by combinations of multiple attributes. Addressing intersectional fairness requires sufficient sample sizes within these subgroups, which was not feasible in this study due to the limited representation of certain intersectional subgroups in the ABIDE II and ADNI datasets.

To address this limitation, future extensions of MDANN could incorporate techniques to evaluate and mitigate intersectional biases. One potential direction is the development of subgroup-specific adversarial components tailored to specific intersections of sensitive features. These components could collaboratively address compounding biases in the data, improving fairness across intersecting subgroups. Additionally, hierarchical modeling or fairness constraints optimized for intersectional subgroups offer promising avenues for future work. Incorporating such techniques would make MDANN more robust to nuanced fairness concerns and better suited for complex, real-world applications.

Nonetheless, there are future opportunities for extending and enhancing MDANN. First, we acknowledge that further improvement in interpretability should be explored within the framework of MDANN. Integrating attention mechanisms and feature attribution methods (e.g., Grad-CAM [30], LIME [31], SHAP [32]) could enhance transparency in the decision-making process and provide insights into which features contribute most to the predictions. These methods could make MDANN more interpretable for clinicians, aligning it with the growing need for explainable AI (XAI) in medical applications. Second, it should be recognized that the current structure of adversarial components is still simple and uniform in MDANN. We will extend MDANN by integrating adversarial components with distinct architectures. Such an approach would enable the training of specialized components tailored to each sensitive feature, accommodating its unique distribution characteristics.

Further, we plan to incorporate more advanced techniques to handle data imbalance and bias. These could include incorporating improved optimization strategies or algorithmic approaches that target specific fairness concerns, as well as exploring more complex feature extraction models beyond the current convolutional autoencoder. For example, variations of autoencoders (such as variational autoencoders) or other novel deep learning architectures like transformers [33] could be integrated into MDANN to enhance its robustness and improve predictive accuracy across a broader range of disease prediction tasks.

## Supporting information

**S1 Table. Class-stratified demographics for the ADNI dataset across sensitive attributes.** For the ADNI dataset, the majority group corresponds to individuals over 78 years old, with more than 18 years of education, and are right-handed.
(PDF)

**S2 Table. Class-stratified demographics for the autism dataset across sensitive attributes.** For the Autism dataset, the majority group corresponds to the right-handedness and male subpopulation.
(PDF)

## Author contributions

**Conceptualization:** Bin Li, Xiaoqian Jiang, Kai Zhang, Arif O Harmanci, Hongchang Gao, Xinghua Shi.

**Methodology:** Bin Li, Hongchang Gao, Xinghua Shi.

**Resources:** Hongchang Gao.

**Software:** Bin Li.

**Supervision:** Xinghua Shi.

**Writing – original draft:** Bin Li.

**Writing – review & editing:** Xiaoqian Jiang, Kai Zhang, Arif O Harmanci, Bradley Malin, Hongchang Gao, Xinghua Shi.

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
