## [Decision Letter · Decision Letter 0]

29 Jul 2024

PDIG-D-24-00118

Enhancing Fairness in Disease Prediction by Optimizing Multiple Domain Adversarial Networks

PLOS Digital Health

Dear Dr. Shi,

Thank you for submitting your manuscript to PLOS Digital Health. After careful consideration, we feel that it has merit but does not fully meet PLOS Digital Health's publication criteria as it currently stands. Therefore, we invite you to submit a revised version of the manuscript that addresses the points raised during the review process.

Please submit your revised manuscript within 60 days Sep 27 2024 11:59PM. If you will need more time than this to complete your revisions, please reply to this message or contact the journal office at digitalhealth@plos.org. Please include the following items when submitting your revised manuscript:

We look forward to receiving your revised manuscript.

Kind regards,

Gloria Hyunjung Kwak

Section Editor

PLOS Digital Health

Journal Requirements:

Additional Editor Comments (if provided):

Reviewers' comments:

Reviewer's Responses to Questions

**Comments to the Author**

1. Does this manuscript meet PLOS Digital Health’s publication criteria? Is the manuscript technically sound, and do the data support the conclusions? The manuscript must describe methodologically and ethically rigorous research with conclusions that are appropriately drawn based on the data presented.

Reviewer #1: Yes

Reviewer #2: Yes

2. Has the statistical analysis been performed appropriately and rigorously?

Reviewer #1: Yes

Reviewer #2: I don't know

3. Have the authors made all data underlying the findings in their manuscript fully available (please refer to the Data Availability Statement at the start of the manuscript PDF file)?

Reviewer #1: No

Reviewer #2: Yes

4. Is the manuscript presented in an intelligible fashion and written in standard English?

PLOS Digital Health does not copyedit accepted manuscripts, so the language in submitted articles must be clear, correct, and unambiguous. Any typographical or grammatical errors should be corrected at revision, so please note any specific errors here.

Reviewer #1: Yes

Reviewer #2: Yes

5. Review Comments to the Author

Please use the space provided to explain your answers to the questions above. You may also include additional comments for the author, including concerns about dual publication, research ethics, or publication ethics. (Please upload your review as an attachment if it exceeds 20,000 characters)

Reviewer #1: Very good study that addresses an important topic. Some suggestions for improvement:

In the introduction, clarify how MDANN can help mitigate bias.

In the method, justify why each experiment was run five times.

The results show still a low accuracy Could this be linked to the low sample size?

Comment in the conclusion whether this could be transferable to other diseases/

Reviewer #2: To achieve fair and accurate disease prediction in biomedicine, this study proposes a novel framework using a Multiple Domain Adversarial Neural Network (MDANN). MDANN architecture tackles bias by incorporating multiple adversarial components. These components learn unbiased patterns through a training process that discourages the model from relying on sensitive features like race or gender. Additionally, pre-trained convolutional autoencoders (CAEs) are integrated to boost both accuracy and fairness. This combined approach aims to deliver reliable disease prediction that is fair and equitable for all patients.

While the proposed framework shows promise, there are several limitations. 

My main concern is the Feature Extractor. CAEs are feature extractors that provide data representations or embeddings in the latent space but do not guarantee the interpretability of the features. Therefore, the quality of the entire system is based on the assumption that the features extracted are correct, belong to the definitions of minority/majority, and are interpretable. Moreover, the choice of adversarial layers is based on the extracted and considered bias features. This implies that the effectiveness of the framework is dependent on the accurate identification and representation of these bias features. 

Another concern is the use of the Loss Function based on the AUC. Particularly for class imbalance, existing solutions, such as those in the Keras library, account for the number of samples per class and incorporate this into their loss functions. These traditional loss functions, often weighted by the number of samples per class, are designed to address the imbalance by giving more importance to the minority class during training. This approach ensures that the model does not become biased towards the majority class, which typically has more samples.

In contrast, the proposed AUC-based loss function aims to enhance the overall classification performance by focusing on the area under the Receiver Operating Characteristic (ROC) curve. While this method can be effective in certain scenarios, it does not inherently account for class imbalance in the same way that sample-weighted loss functions do.

As the authors mentioned (rows 185-186), the AUC-based loss function does not inherently provide a measure of fairness. 

A comparison between the proposed AUC-based loss function and traditional loss functions that are weighted based on the number of samples per class is missing from the current study. This comparison is crucial to demonstrate the effectiveness and advantages of the proposed method in handling class imbalance. Without this comparison, it is difficult to determine whether the proposed approach offers a significant improvement over existing methods, particularly in scenarios with highly imbalanced datasets. 

Finally, the use of deep learning techniques, such as convolutional autoencoders (CAEs) and adversarial networks, may reduce the interpretability of the model, making it harder for clinicians to understand and trust the predictions. While these advanced techniques can enhance prediction accuracy and fairness, their complex nature can obscure how specific features contribute to the model's decisions. This lack of transparency can be a significant barrier to the adoption of such models in clinical practice, where understanding the rationale behind predictions is crucial for informed decision-making and patient trust.

Minor Concerns:

1. The acronym CAE is not defined in the introduction.

2. The definitions of "majority" and "minority" are missing in Equations 2 and 3, which may lead to ambiguity in interpreting the equations.

3. Table 1 should report the F1-score instead of accuracy. When dealing with imbalanced datasets, accuracy is not a reliable metric, and the F1-score provides a better measure of the model's performance.

4. The quality of the figures should be improved, for example by using PDF format or higher resolution images to ensure clarity and readability.

6. PLOS authors have the option to publish the peer review history of their article (what does this mean?). If published, this will include your full peer review and any attached files.

**Do you want your identity to be public for this peer review?** For information about this choice, including consent withdrawal, please see our Privacy Policy.

Reviewer #1: No

Reviewer #2: Yes: Gaetano Manzo

---

## [Decision Letter · Decision Letter 1]

7 Nov 2024

PDIG-D-24-00118R1Enhancing Fairness in Disease Prediction by Optimizing Multiple Domain Adversarial NetworksPLOS Digital Health Dear Dr. Shi, Thank you for submitting your manuscript to PLOS Digital Health. After careful consideration, we feel that it has merit but does not fully meet PLOS Digital Health's publication criteria as it currently stands. Therefore, we invite you to submit a revised version of the manuscript that addresses the points raised during the review process. Please submit your revised manuscript within 60 days Jan 06 2025 11:59PM. If you will need more time than this to complete your revisions, please reply to this message or contact the journal office at digitalhealth@plos.org. Please include the following items when submitting your revised manuscript:* A rebuttal letter that responds to each point raised by the editor and reviewer(s). You should upload this letter as a separate file labeled 'Response to Reviewers'. This file does not need to include responses to any formatting updates and technical items listed in the 'Journal Requirements' section below.* A marked-up copy of your manuscript that highlights changes made to the original version. You should upload this as a separate file labeled 'Revised Manuscript with Track Changes'.* An unmarked version of your revised paper without tracked changes. You should upload this as a separate file labeled 'Manuscript'. If you would like to make changes to your financial disclosure, competing interests statement, or data availability statement, please make these updates within the submission form at the time of resubmission. Guidelines for resubmitting your figure files are available below the reviewer comments at the end of this letter. We look forward to receiving your revised manuscript. Kind regards, Po-Chih Kuo, Ph. D.Section EditorPLOS Digital Health Leo Anthony CeliEditor-in-ChiefPLOS Digital Healthorcid.org/0000-0001-6712-6626 **Journal Requirements:** **Additional Editor Comments (if provided):****Reviewers' Comments:** Reviewer's Responses to Questions

**Comments to the Author**

1. If the authors have adequately addressed your comments raised in a previous round of review and you feel that this manuscript is now acceptable for publication, you may indicate that here to bypass the “Comments to the Author” section, enter your conflict of interest statement in the “Confidential to Editor” section, and submit your "Accept" recommendation.

Reviewer #2: (No Response)

Reviewer #3: (No Response)

2. Does this manuscript meet PLOS Digital Health’s publication criteria? Is the manuscript technically sound, and do the data support the conclusions? The manuscript must describe methodologically and ethically rigorous research with conclusions that are appropriately drawn based on the data presented.

Reviewer #2: Partly

Reviewer #3: No

3. Has the statistical analysis been performed appropriately and rigorously?

Reviewer #2: I don't know

Reviewer #3: No

4. Have the authors made all data underlying the findings in their manuscript fully available (please refer to the Data Availability Statement at the start of the manuscript PDF file)?

Reviewer #2: Yes

Reviewer #3: Yes

5. Is the manuscript presented in an intelligible fashion and written in standard English?

Reviewer #2: Yes

Reviewer #3: Yes

6. Review Comments to the Author

Reviewer #2: The manuscript proposes a framework utilizing a Multiple Domain Adversarial Neural Network (MDANN) to mitigate biases in predictive models for biomedical applications. While the motivation is clear and relevant given the critical need for fairness in medical AI, the paper has several significant issues that still need to be addressed:

Firstly, although the authors claim that their MDANN framework mitigates multiple biases across sensitive features, the fairness component is not rigorously addressed. The use of convolutional autoencoders (CAEs) to extract "deep representations" is problematic because CAEs are designed to preserve the essential components of the data. If the biased feature is central to the image (e.g., demographic traits), the CAE will likely retain it. This fundamentally contradicts the claim of bias mitigation. The authors need to provide a thorough analysis or evidence showing how their model effectively disentangles biased components from the essential features. Without this, the assertion of fairness remains unsubstantiated.

Secondly, the manuscript employs the Area Under the Curve (AUC) as a loss function to address class imbalance, but the reasoning behind this choice is not sufficiently justified. AUC-based optimization can be unreliable in practice, particularly with imbalanced biomedical data, where models may fail to properly learn minority class distinctions. A more standard approach would be weighted cross-entropy loss, which directly addresses imbalance by adjusting for class frequencies. A comparative analysis between AUC-based and weighted cross-entropy approaches is missing, leaving the reader unclear about why the proposed method is preferable and how it truly impacts fairness and classification performance.

The use of accuracy as a primary metric to evaluate model performance is inappropriate for imbalanced and biased datasets, such as those encountered in biomedical applications. Accuracy can be misleading, particularly in scenarios where one class dominates the dataset, as it does not capture the model’s ability to correctly predict the minority class or address fairness across sensitive features. The authors should provide metrics like F1-score, precision, recall, or balanced accuracy, which are more appropriate for evaluating the effectiveness of their model in mitigating bias and ensuring equitable outcomes.

Additionally, there are formatting issues in the manuscript that hinder readability. For instance, the caption for Figure 1 is floating separately from the figure itself, creating confusion for readers trying to follow the explanations. Correct placement of figures is essential for proper interpretation, and this issue must be addressed to improve the clarity and flow of the manuscript.

Reviewer #3: 1) Experimental setup: The author only mentioned that the each experiment was run five times and averaged results are reported, without any specific details about cross-validation. Please explicit specify the details regarding the data parition or cross-validation, including the type of data partition/cross-validation method used (e.g. K-fold cross-validation, bootstrap cross-validation, or a single holdout testing test) and their specific details (e.g. ratio of training/testing paritions, or whether the partiions were stratified by class or sensitive attributes).

2) On the equations of fairness metrics:

a) Y is not defined in the manuscript. In the context of this study, Y the ground truths, or the model predictions?

b) “the proportion of the minority group that received 231 the positive outcome”: what is the “positive outcome” specifically in detecting Alzheimer’s Disease and Autism?

c) The equation for EO is incorrect. In order to estimate true positive rate, both ground truth and model prediction should be present in this equation. However only “Y” was included (which is also not clear whether it denotes the ground truths of the predictions).

d) For disparity in recalls, the authors only reported the disparity in true positive rate (TPR). However, only reporting TPR disparity does not capture the whole picture of the fairness performance trade-off (an extreme example being both demographics group both having TPRs of 0, which is still considered perfectly fair if only disparity for TPR were reported). For a more comprehansive evaluation, please report the disparity of true negative rate, and the performance metrics that which the fairness metrics are derived from (DI: selection rate; EO: TPR; TNR disparity: TNR)

3) Line 334-335: “panel (a) 334 illustrates the accuracy and AUC score of predictors employing different numbers of adversarial components while incorporating cross-entropy loss into the MDANN.” , and line 344-345: “Fig. 5 (b) illustrates the accuracy and AUC score when employing minimax loss for training. “. These descriptions do not match the figure caption for Fig. 5. According to the caption it should be panel A-B and panel C-D respectively.

4) For the performance metrics and fairness metrics reported in this manuscript, please also report their variation (e.g. standard deviation) or associated estimates of uncertainty (e.g. confidence intervals).

5) Caption for Fig. 6: For better clarity, please specify what classification task is used in this figure. Same for the captions of other figures & tables.

6) On the description for Fig. 6: EO was not reported in the main text. Please report all performance/fairness metrics described in this study in either main text, tables or supplementary tables, for each of the tasks, and models compared.

7) On subsection “Bias mitigation via multiple components in the adversarial module”: It comes to my attention that no fairness metric was reported for AD prediction in this subsection (Fig. 5 only includes performance metrics). As a result, there is no quantitative evidence that draws to the conclusion that “fairness was significantly improved without sacrificing classification performance”. Same the previous comment, please provide comprehensive quantitative results for both of the classification tasks.

8) The authors mentioned in the introduction that “biases can intersect in complex ways that are difficult to disentangle and address with single-attribute-focused methods”. However, it should be noted that the methodology and evaluation in this study is still essentially based on marginal fairness, i.e. fairness defined by single sensitive attributes (although their model can mitigate biases for each of the attribute simultaneously, the adversarial modules still operates independently without considering the interactions between sensitive attribtues). Although marginal fairness could be used to approximate intersectional fairness, there is still a distinction between these two fairness definitions, and marginal fairness on all individual attributes does not quarantee fairness in intersectional subgroups defined by multiple attributes. That being said, it would be benificial if the author could also evaluate fairness across intersectional subgroups defined by multiple attributes (if it is feasible for the sample sizes in this study). Or, in the very least, the authors should clarify that their method assumes no bias interaction between sensitive attributes.

9) Fig. 3 only provide a partial view of the demographic distribution, since we cannot tell if there is any difference in class distribution between patient groups. Please consider also report class-stratified demographics. In addition, please also provide specific patient counts in addition to percentages.

7. PLOS authors have the option to publish the peer review history of their article (what does this mean?). If published, this will include your full peer review and any attached files.

**Do you want your identity to be public for this peer review?** For information about this choice, including consent withdrawal, please see our Privacy Policy.

Reviewer #2: **Yes: **Gaetano Manzo

Reviewer #3: **Yes: **Shih-Yen Lin

---

## [Decision Letter · Decision Letter 2]

10 Feb 2025

PDIG-D-24-00118R2Enhancing Fairness in Disease Prediction by Optimizing Multiple Domain Adversarial NetworksPLOS Digital Health Dear Dr. Shi, Thank you for submitting your manuscript to PLOS Digital Health. After careful consideration, we feel that it has merit but does not fully meet PLOS Digital Health's publication criteria as it currently stands. Therefore, we invite you to submit a revised version of the manuscript that addresses the points raised during the review process. Please submit your revised manuscript within 30 days Mar 12 2025 11:59PM. If you will need more time than this to complete your revisions, please reply to this message or contact the journal office at digitalhealth@plos.org. Please include the following items when submitting your revised manuscript:* A rebuttal letter that responds to each point raised by the editor and reviewer(s). You should upload this letter as a separate file labeled 'Response to Reviewers'. This file does not need to include responses to any formatting updates and technical items listed in the 'Journal Requirements' section below.* A marked-up copy of your manuscript that highlights changes made to the original version. You should upload this as a separate file labeled 'Revised Manuscript with Track Changes'.* An unmarked version of your revised paper without tracked changes. You should upload this as a separate file labeled 'Manuscript'. If you would like to make changes to your financial disclosure, competing interests statement, or data availability statement, please make these updates within the submission form at the time of resubmission. Guidelines for resubmitting your figure files are available below the reviewer comments at the end of this letter. We look forward to receiving your revised manuscript. Kind regards, Po-Chih Kuo, Ph. D.Section EditorPLOS Digital Health Po-Chih KuoSection EditorPLOS Digital Health Leo Anthony CeliEditor-in-ChiefPLOS Digital Healthorcid.org/0000-0001-6712-6626  **Additional Editor Comments (if provided):****Reviewers' Comments:** Reviewer's Responses to Questions

**Comments to the Author**

1. If the authors have adequately addressed your comments raised in a previous round of review and you feel that this manuscript is now acceptable for publication, you may indicate that here to bypass the “Comments to the Author” section, enter your conflict of interest statement in the “Confidential to Editor” section, and submit your "Accept" recommendation.

Reviewer #3: (No Response)

2. Does this manuscript meet PLOS Digital Health’s publication criteria? Is the manuscript technically sound, and do the data support the conclusions? The manuscript must describe methodologically and ethically rigorous research with conclusions that are appropriately drawn based on the data presented.

Reviewer #3: Yes

3. Has the statistical analysis been performed appropriately and rigorously?

Reviewer #3: Yes

4. Have the authors made all data underlying the findings in their manuscript fully available (please refer to the Data Availability Statement at the start of the manuscript PDF file)?

Reviewer #3: Yes

5. Is the manuscript presented in an intelligible fashion and written in standard English?

Reviewer #3: Yes

6. Review Comments to the Author

Reviewer #3: 1) The authors claimed to have added TNRD to the manuscript. But I could not find any specific value of TNRD throughout their revised manuscript.

2) On the response to previous comments, “Accuracy is inappropriate for imbalanced and biased datasets. Metrics like F1-score, precision, recall, or balanced accuracy should be included. (from Comment 3 from Reviewer 2)"; and “For a more comprehensive evaluation, please report the disparity of true negative rate, and the performance metrics that which the fairness metrics are derived from (DI: selection rate; EO: TPR; TNR disparity: TNR) (from Comment 2d from Reviewer 3)": The authors do not seem to have fully addressed these comments. They only included F1-score in Table 1 (which is only an ablation study for CAEs), while the performance metrics mentioned above are still missing from the rest of the manuscript. The only two performance metrics that were consistently reported throughout the manuscript are accuracy and AUROC, which - as mentioned in the previous comments - do not fully capture the model performance and fairness-performance trade-off.

7. PLOS authors have the option to publish the peer review history of their article (what does this mean?). If published, this will include your full peer review and any attached files.

**Do you want your identity to be public for this peer review?** For information about this choice, including consent withdrawal, please see our Privacy Policy.

Reviewer #3: No

---

## [Decision Letter · Decision Letter 3]

19 Mar 2025

Enhancing Fairness in Disease Prediction by Optimizing Multiple Domain Adversarial Networks

PDIG-D-24-00118R3

Dear Associate Professor Shi,

We are pleased to inform you that your manuscript 'Enhancing Fairness in Disease Prediction by Optimizing Multiple Domain Adversarial Networks' has been provisionally accepted for publication in PLOS Digital Health.

Best regards,

Po-Chih Kuo, Ph. D.

Section Editor

PLOS Digital Health

**Additional Editor Comments (if provided):**

**Reviewer Comments (if any, and for reference):**

Reviewer's Responses to Questions

**Comments to the Author**

1. If the authors have adequately addressed your comments raised in a previous round of review and you feel that this manuscript is now acceptable for publication, you may indicate that here to bypass the “Comments to the Author” section, enter your conflict of interest statement in the “Confidential to Editor” section, and submit your "Accept" recommendation.

Reviewer #3: All comments have been addressed

2. Does this manuscript meet PLOS Digital Health’s publication criteria? Is the manuscript technically sound, and do the data support the conclusions? The manuscript must describe methodologically and ethically rigorous research with conclusions that are appropriately drawn based on the data presented.

Reviewer #3: Yes

3. Has the statistical analysis been performed appropriately and rigorously?

Reviewer #3: Yes

4. Have the authors made all data underlying the findings in their manuscript fully available (please refer to the Data Availability Statement at the start of the manuscript PDF file)?

Reviewer #3: Yes

5. Is the manuscript presented in an intelligible fashion and written in standard English?

Reviewer #3: Yes

6. Review Comments to the Author

Reviewer #3: The authors have addressed all of the comments, and the manuscript has met the publication criteria.

7. PLOS authors have the option to publish the peer review history of their article (what does this mean?). If published, this will include your full peer review and any attached files.

**Do you want your identity to be public for this peer review?** For information about this choice, including consent withdrawal, please see our Privacy Policy.

Reviewer #3: **Yes: **Shih-Yen Lin
